# Chromatographic Scalable Method to Isolate Engineered Extracellular Vesicles Derived from Mesenchymal Stem Cells for the Treatment of Liver Fibrosis in Mice

**DOI:** 10.3390/ijms24119586

**Published:** 2023-05-31

**Authors:** Luciana M. Domínguez, Bárbara Bueloni, Ma. José Cantero, Milagros Albornoz, Natalia Pacienza, Celeste Biani, Carlos Luzzani, Santiago Miriuka, Mariana García, Catalina Atorrasagasti, Gustavo Yannarelli, Juan Bayo, Esteban Fiore, Guillermo Mazzolini

**Affiliations:** 1Laboratorio de Terapia Génica, Instituto de Investigaciones en Medicina Traslacional (IIMT), Universidad Austral—CONICET, Pilar B1629, Buenos Aires, Argentinacanter.maj@gmail.com (M.J.C.); mggarcia24@gmail.com (M.G.); mcatalinaa@gmail.com (C.A.); jmbayo@hotmail.com (J.B.); 2Instituto de Medicina Traslacional, Trasplante y Bioingeniería (IMeTTyB), Universidad Favaloro-CONICET, Ciudad Autónoma de Buenos Aires C1078, Argentina; npacienza@favaloro.edu.ar (N.P.);; 3LIAN-CONICET, Fleni, Belén de Escobar B1625, Buenos Aires, Argentina; 4Liver Unit, Hospital Universitario Austral, Universidad Austral—CONICET, Pilar B1629, Buenos Aires, Argentina

**Keywords:** liver fibrosis, extracellular vesicles, mesenchymal stromal cells, chromatography EVs isolation, engineered EVs

## Abstract

New therapeutic options for liver cirrhosis are needed. Mesenchymal stem cell (MSC)-derived extracellular vesicles (EVs) have emerged as a promising tool for delivering therapeutic factors in regenerative medicine. Our aim is to establish a new therapeutic tool that employs EVs derived from MSCs to deliver therapeutic factors for liver fibrosis. EVs were isolated from supernatants of adipose tissue MSCs, induced-pluripotent-stem-cell-derived MSCs, and umbilical cord perivascular cells (HUCPVC-EVs) by ion exchange chromatography (IEC). To produce engineered EVs, HUCPVCs were transduced with adenoviruses that code for insulin-like growth factor 1 (AdhIGF-I-HUCPVC-EVs) or green fluorescent protein. EVs were characterized by electron microscopy, flow cytometry, ELISA, and proteomic analysis. We evaluated EVs’ antifibrotic effect in thioacetamide-induced liver fibrosis in mice and on hepatic stellate cells in vitro. We found that IEC-isolated HUCPVC-EVs have an analogous phenotype and antifibrotic activity to those isolated by ultracentrifugation. EVs derived from the three MSCs sources showed a similar phenotype and antifibrotic potential. EVs derived from AdhIGF-I-HUCPVC carried IGF-1 and showed a higher therapeutic effect in vitro and in vivo. Remarkably, proteomic analysis revealed that HUCPVC-EVs carry key proteins involved in their antifibrotic process. This scalable MSC-derived EV manufacturing strategy is a promising therapeutic tool for liver fibrosis.

## 1. Introduction

Cirrhosis, the end stage of liver fibrosis, is characterized by extracellular matrix accumulation (ECM) and liver function impairment [1,2]. Typically, chronic inflammation and cell death in the liver trigger the activation of hepatic stellate cells (HeSCs). These cells proliferate, migrate to the injured area, and secrete large amounts of ECM, mainly collagen type I and III, and pro-fibrogenic factors, which ultimately results in fibrosis and cirrhosis [1]. Currently, a liver transplant is the only curative option for end-stage liver disease, but the shortage of organ donors and the mortality on the waiting list highlights the need for new therapeutics options. In recent years, mesenchymal stem/stromal cells (MSCs) have shown promise as a potential therapy for cirrhosis due to their ability to modulate liver inflammation and promote regeneration [3,4,5,6,7].

MSCs are immune-privileged multipotent cells able to regulate the immune/inflammatory response [7,8]. These cells are phenotypically and functionality described according to the International Society for Cellular Therapy (ISCT): MSCs must be plastic-adherent when maintained in standard culture conditions, express CD105, CD73, and CD90, and lack the expression of CD45, CD34, CD14 or CD11b, CD79alpha or CD19, and HLA-DR surface molecules. Moreover, MSCs must differentiate to osteoblasts, adipocytes and chondroblasts in vitro [6].

MSCs have been isolated from diverse human tissues, but clinically relevant sources for cell therapy are the bone marrow, adipose tissue, and birth-associated tissue including the placenta, amnion, and umbilical cord [9,10]. In particular, human umbilical cord perivascular cells (HUCPVCs) are a valued source of MSCs for cell therapy due to their accessibility, faster doubling time, and low variability on growth kinetics between donors [11,12]. In addition, induced pluripotent stem cells (iPSCs) have emerged as a novel source to obtain MSCs in quantity and quality for their clinical use, keeping their anti-inflammatory and pro-regenerative characteristics [13,14]. However, the transference of these MSC-derived experimental treatments to the clinic has presented several obstacles due the risk of lung embolism and lack of quality standards required for their use in humans [15,16]. In the last years, extracellular vesicles (EVs) derived from MSCs emerged as an attractive alternative to cell-based therapies due their potential to mimic the therapeutic effects of their cells of origin, avoiding the risks associated with the use of cells.

EVs are membrane-covered vesicles involved in cell–cell communication through the delivery of molecular cargos such as proteins, lipids, messenger RNAs (mRNAs), and microRNAs (miRNAs) [17,18,19]. In particular, exosomes are a subset of very small EVs (~40–100 nm) characterized by the presence of tetraspanins (CD9, CD63, and CD81) that are originated in the endosomal compartment and secreted into the extracellular space when multivesicular endosomes are fused with the cell membrane [19]. In addition, EVs show a low toxicity and lung embolization risk in vivo, are poorly immunogenic, are stable for long periods, can be stored at −80 °C, and are engineered to generate EVs that can deliver specific molecular cargos [3,20,21,22,23]. Furthermore, MSC-derived EVs have been used for the treatment of inflammatory and degenerative diseases including liver fibrosis in animal models [24,25].

A key point in the development of therapeutic strategies based on MSC-derived EVs is to establish an efficient, fast, and scalable isolation method. In addition, when MSCs are engineered to deliver specific molecular cargos on their EVs, the chosen method must conserve its quality, composition, and biological effect. Recently, we demonstrated that the anti-fibrotic effect of HUCPVCs transduced with adenovirus to over-express insulin-like growth factor 1 (IGF-I) (AdhIGFI-HUCPVCs) is mediated by EVs. Remarkably, AdhIGF-I-HUCPVC-derived EVs isolated by an ultracentrifugation method are able to transport IGF-1 and recapitulate the therapeutic effect of cell transplantation [25].

The aim of this study was to develop a new tool using EVs derived from MSCs to deliver therapeutic factors for liver fibrosis treatment. First, we compare a scalable method based on ion exchange chromatography [26] for isolating EVs from HUCPVCs with the classic ultracentrifugation method. Second, we evaluated the antifibrotic therapeutic potential of EVs derived from different clinically relevant sources of MSCs: adipose tissue (ASC-EVs), HUCPVC-EVs, and induced-pluripotent-stem-cell-derived MSCs (iMSC-EVs). Finally, we confirmed that chromatograph isolation is a suitable method for the purification of engineered EVs, keeping their potential to carry IGF-1 and their in vivo therapeutic effect on liver fibrosis in mice.

## 2. Results

### 2.1. EVs Derived from HUCPVCs Isolated by Ion Exchange Chromatography Retain Their Typical Characteristics

In our previous work we demonstrated that EVs isolated by ultracentrifugation are at least in part responsible for the antifibrotic effects of HUCPVCs. However, the method employed for EV purification has limited scale-up potential [25]. Therefore, to establish a protocol to produce EVs derived from MSCs with clinical potential, we compared EVs isolated by ultracentrifugation with those produced by a scalable method based on ion exchange chromatography (IEC) (Figure 1A).

The presence of EVs was first assessed by protein quantification in the pellet after centrifugation and in the eight fractions obtained from the chromatography elution. As shown in Figure 1B, high levels of proteins were detected both in the ultracentrifugation pellet and in the fractions #3, #4, and #5 obtained from chromatography. In addition, particle size distribution and number were assessed by Microfluidics Resistive Pulse Sensing (MRPS). In agreement with the protein quantification data, in the same three elution fractions, a high number of particles with a similar size distribution profile as EVs isolated by ultracentrifugation were detected (Figure 1C,D). Then, by transmission electron microscopy, we confirmed the presence of nanoparticles that preserve the characteristic shape and size of EVs in the pool of isolated fractions by chromatography (Figure 1C) and ultracentrifugation pellet (Figure 1D). Furthermore, we used beads coated with anti-CD63 antibody to trap EVs and then assessed CD9 and CD81 markers by standard flow cytometry. As expected, the expression levels of EV markers in the three elution fractions were similar to those observed in the pellet obtained by ultracentrifugation (Figure 1E).

To evaluate whether EVs maintain their biological activity in both isolation methods, activated HeSC (CFSC-2G cell line) were incubated with the elution fraction #3, #4, and #5; a pool of these elution fractions; and EVs isolated by ultracentrifugation. Then, gene expression of the hepatic stellate cell activation markers collagen type 1A2 (COL1A2) and alpha-smooth muscle actin (α-SMA) were assessed. As expected, the expression of both genes was downregulated in HeSCs treated with each elution fraction, the pool, and ultracentrifugation EVs (Figure 1H,I). These results confirmed that HUCPVC-derived EVs isolated by IEC retain the typical characteristics of EVs and preserve their anti-fibrotic effect.

### 2.2. EVs Derived from MSCs from Adipose Tissue, Umbilical Cord, and iMSCs Share Similar Phenotypic Characteristics

To gain insight into the development of MSC-derived EVs as a therapeutic tool for liver fibrosis, we decided to compare the therapeutic potential of EVs derived from different clinically relevant sources of MSCs. EVs were isolated by IEC from supernatants of adipose-tissue-derived MSCs (ASC-EVs), induced-pluripotent-stem-cell-derived MSCs (iMSC-EVs), and umbilical cord perivascular cells (HUCPVC-EVs) (Figure 2A). EV isolation was first evaluated by protein quantification on elution fractions, which shows the highest levels in fractions #4, #5, and #6 of each MSC source (Figure 2B–D). Then, the presence of EVs was confirmed on pooled fractions by flow cytometry. As shown in Figure 2E, EVs captured with the CD63 antibody of three MSC sources were positive for CD81 (~95%). Nevertheless, while iMSC-EVs and HUCPVC-EVs were positive for CD9 (62% and 68%, respectively), ASC-EVs were negative (Figure 2E). Moreover, the MRPS analysis of pooled fractions showed a similar profile on the size distribution and particle concentration of the three MSC sources (Figure 2F). Altogether, we found that EVs derived from ASCs, HUCPVCs, and iMSCs isolated by IEC present similar phenotypic characteristics.

### 2.3. Treatment with EVs Derived from the Three MSCs Sources Ameliorates Liver Fibrosis and Induces Hepatic Regeneration

To further explore the therapeutic potential of EVs derived from different MSC sources, we assessed the anti-fibrotic effect of ASC-EVs, iMSC-EVs, and HUCPVC-EVs in the experimental mice model of liver fibrosis induced by TAA administration. On week 6, EVs were i.v. injected every 5 days for a total of three doses. At week 8, animals were euthanized and liver samples analyzed, as shown in the schematic of Figure 3A. Liver fibrosis was analyzed by Sirius Red staining of collagen deposits on the liver section and quantified by morphometry of the positive areas. It should be noted that systemic administration of ASC-EVs, iMSC-EVs, and HUCPVC-EVs reduced collagen deposits in comparison with the saline group (Figure 3B,C). In addition, HSC activation was analyzed in vivo on liver tissue by immunostaining for α-SMA. Consistent with the above results, the positive stained area for α-SMA decreased in liver sections from ASC-EV-, iMSC-EV-, and HUCPVC-EV-treated mice when compared with saline (Figure 3D,E). Then, we confirmed these results by a real-time qPCR of liver samples. As expected, expression levels of COL1A2 and α- SMA mRNA were lower in the mice treated with ASC-EVs, iMSC-EVs, and HUCPVC-EVs in comparison with saline (Figure 3H,I).

To evaluate the regenerative capacity of MSC-derived EVs, we analyzed the expression levels of the proliferation marker PCNA on liver tissue. Remarkably, PCNA-positive cells were found to be significantly increased after the application of ASC-EVs, iMSC-EVs, and HUCPVC-EVs compared with the saline group (Figure 3F,G), indicating that EVs from different sources induce hepatocyte proliferation and might promote liver regeneration.

Considering the anti-fibrogenic potential of the different EVs, we decide to compare the effect in vitro of MSC-derived EVs on HeSC activation status. As described above, CFSC-2G cells were incubated with the EVs derived from the different MSC sources or DMEM as control, and fibrogenic genes were analyzed. As observed in Figure 3J,K, incubation of CFSC-2G with EVs from the three MSCs sources down-regulated, in a similar amount, the gene expression of COL1A2 (*p* < 0.01 vs. DMEM) and α-SMA (*p* < 0.01 vs. DMEM).

In view of the similar anti-fibrotic and pro-regenerative therapeutic potential of the three MSC-EVs, we decided to evaluate the use of HUCPVC-EVs for the delivery of therapeutics factors to treat liver fibrosis.

### 2.4. EVs Derived from AdhIGFI-HUCPVCs Isolated by Ion Exchange Chromatography Are Loaded with IGF-I

Considering the above results, we decided to determine if the IEC purification method affects engineered EVs. Previously, we demonstrated that EVs engineered to load and transport IGF-1 can be isolated by ultracentrifugation from the conditioned media of AdhIGFI-HUCPVCs [25]. Here, we analyzed whether the isolation method by IEC affects the quality of AdhIGFI-HUCPVC-derived EVs. First, to optimize the IGF-I production and loading on EVs, we set up the infection conditions of HUCPVCs with adenoviruses carrying IGFI and green fluorescent protein (GFP) as a control. HUCPVCs were infected at different multiplicity of infection (MOIs) (1 to 30) of AdhIGFI and AdGFP vectors. After 2 days of infection, the IGF-I production was analyzed by ELISA in a conditioned media (CM) at different MOIs. A significant increase in IGF-I levels was found in the supernatant of AdhIGFI-HUCPVCs when compared with AdGFP-HUCPVCs (Appendix A). In addition, the viability of HUCPVCs was reduced when MOIs higher than 10 were used (Appendix A). Next, to determine if the infection could affect the anti-inflammatory capability of the HUCPVCs, we analyzed the TNF-α secretion by LPS-activated J774.1 macrophages co-incubated with CM of HUCPVCs infected at different MOIs with AdhIGF-I and AdGFP. As shown in Appendix A, the TNF-α levels were reduced in all infection conditions in similar levels without significant changes in comparison with non-infected HUCPVCs. Therefore, we selected the MOI of 10 for the following experiments.

Then, EVs were isolated by IEC from the supernatant of AdhIGFI-HUCPVCs. As described in the previous section, high protein levels were concentrated at fractions #4, #5, and #6 (Figure 4A). In addition, the presence of IGF-I was measured in each fraction by ELISA. Consistently with the higher protein content of fractions #4, #5, and #6, the highest levels of IGF-I were detected in the fractions #3 to #6 (Figure 4B). Moreover, the EVs captured with the CD63 antibody on AdhIGFI-HUCPVC and AdGFP-HUCPVC pools were positive for CD81 (>96%). Nevertheless, AdhIGF-I-HUCPVC-EVs and HUCPVC-EVs were 29% and 68% positive for CD9, respectively (Figure 4C). Additionally, the MRPS analysis showed a slight difference in the size dispersion of AdhIGF-I-HUCPVC-EVs compared to HUCPVC-EVs (Figure 4D).

Finally, we quantified the amount of IGF-I within EVs to confirm that they are effectively loaded after chromatography isolation. To this end, EVs derived from AdhIGFI-HUCPVCs were first dialyzed (300 kDa cut-off) to remove soluble proteins, and then lysed (Figure 4E) to measure the loaded IGF-I. As expected, IGF-I levels in the dialyzed AdhIGFI-HUCPVC-EV preparation were higher after lysis compared to non-lysated EVs. Therefore, since the total IGF-I is present in the lysis without dialysis condition, we deduced that IGF-I can be present on both the inside (dialysis plus lysis condition) and outside (without both dialysis and lysis condition) of the EVs. Even more, high concentrations of IGF-I were detected in the dialyzed EV fraction without lysis, indicating that IGF-I could be bound to the EV outer surface and hence be potentially transported to the target cell (Figure 4E). In summary, these results demonstrate that AdhIGFI-HUCPVC-EVs isolated by chromatography could carry and transport therapeutic factors.

### 2.5. AdhIGFI-HUCPVCs-Derived EVs Isolated by Chromatography Ameliorate Liver Fibrosis and Induce Liver Regeneration

To evaluate if AdhIGFI-HUCPVC-EVs isolated by IEC retain the therapeutic potential against liver fibrosis, we evaluated their anti-fibrotic effect in the TAA experimental model. Treatment was started after 6 weeks of fibrosis induction, and EVs derived from AdhIGFI-HUCPVCs and AdGFP-HUCPVCs were i.v. injected every 5 days for a total of 3 doses. At week 8, animals were euthanized (Figure 5A). As shown in liver sections stained with Sirius Red, the systemic administration of EVs derived from AdhIGFI-HUCPVCs and AdGFP-HUCPVCs reduced collagen deposits (*p* < 0.001). Strikingly, the degree of liver fibrosis in mice treated with AdhIGFI-HUCPVC-derived EVs was significantly lower when compared to EVs derived from AdGFP-HUCPVCs (Figure 5B,C). Consistently, a decrease in the α-SMA stained area was found in sections from AdhIGFI-HUCPVC-EV-treated mice when compared with the controls (Figure 5D,E). In addition, the mRNA expression levels of COL1A2 and α-SMA were also analyzed in liver samples. As expected, expression levels of both pro-fibrogenic markers were significantly downregulated in liver samples obtained from AdhIGFI-HUCPVC-derived-EVs-treated animals when compared to those treated with EVs from HUCPVCs and AdGFP-HUCPVCs (Figure 5H,I).

Moreover, we analyzed the regenerative potential of EVs loaded with IGF-I. As shown in Figure 5F, PCNA-positive hepatocytes were found to be significantly increased after the application of AdhIGFI-HUCPVC-EVs compared with AdGPF-HUCPVC-EVs and the saline group (*p* < 0.0001) (Figure 5F,G).

Finally, the in vitro biological effect of EVs loaded with IGF-I was evaluated on HeSCs activation. CFSC-2G cells were incubated overnight with EVs derived from AdGFP-HUCPVCs, AdhIGFI-HUCPVCs, or DMEM as described above, and fibrogenic genes were analyzed. As observed in Figure 5J,K, mRNA expression levels of COL1A2 and α-SMA were downregulated in HeSCs treated with EVs derived from AdhIGFI-HUCPVCs (*p* < 0.001 vs. DMEM), indicating a reduction of HSC activation status. From the previous data, we can conclude that AdhIGFI-HUCPVC-EVs isolated by a chromatographic method ameliorate liver fibrosis, reduce HeSC activation, and promote liver regeneration.

### 2.6. The Proteome of EVs Derived from HUCPVCs Is Related to Their Anti-Fibrotic Potential

Finally, we performed a proteomic analysis of EVs derived from HUCPVCs, AdGFP-HUCPVCs, and AdhIGFI-HUCPVCs to further characterize their anti-fibrotic effect. As shown in the overlap analysis of identified proteins, the cargo is highly shared among the three types of EVs (Figure 6A and Appendix A). The differential expression analysis between the three types of EVs showed a small number of proteins over-represented on AdhIGF-I-HUCPVC-EVs compared to HUCPVC-EVs (Figure 6B and Appendix A). The gene ontology (GO) analysis (“Molecular Function” and “Biological Process”) revealed that these proteins are involved in terms unrelated to enhanced anti-fibrotic potential (Appendix A). Additionally, no over-represented proteins were detected in AdhIGF-I-HUCPVC-EVs compared to AdGFP-HUCPVC-EVs (Figure 6C and Appendix A).

Taking these results into consideration, we performed a GO and pathway analysis of proteins identified in the three types of EVs. As shown in Figure 6D,E, among the top ten molecular function terms, we found “Peptidase activity” and “Endopeptidase activity” (Figure 6D). In addition, “Tissue regeneration” is one of the top ten biological processes (Figure 6E). Together, these terms could be related to the anti-fibrotic effect observed after the EV treatment. On the other hand, the top ten terms of the “Cellular Components” GO analysis are related to EV biology such as “Secretory granule”, “Cytoplasmic vesicle lumen”, or “Secretory vesicle” (Appendix A). Strikingly, when looking further for anti-fibrotic-relevant biological processes, we found terms such as “Tissue regeneration”, “Wound healing”, “Collagen metabolic process”, “Negative regulation of cell death”, “Regulation of cell migration”, “Negative regulation of Transforming Growth Factor beta (TGF-β) Receptor 1 signaling pathway”, and “Negative regulation of TGF-β production” (Figure 6F). Altogether, these analyses suggest that, in general, EVs derived from HUCPVCs contain proteins involved in their anti-fibrotic process.

In conclusion, our study demonstrates that MSCs-EVs can be efficiently isolated using a scalable anion exchange chromatography method, preserving the quantity, quality, and biological function of EVs. Furthermore, we have shown that EVs derived from engineered MSCs, particularly HUCPVCs, are capable of loading and transporting specific cargos such as IGF-I. Therefore, the chromatographic method is effective in maintaining the therapeutic potential of engineered EVs, making them a promising tool for treating liver fibrosis.

## 3. Discussion

The liver transplant is the only curative treatment for end-stage liver fibrosis, but the shortage of organ donors highlights the urgent need for new therapeutics options [1]. Cell therapy employing MSCs has been extensively explored in pre-clinical and clinical studies as a strategy to avoid or delay liver transplantation [3,27]. MSCs have been isolated from various sources including bone marrow, adipose tissue, and birth-associated tissues such as umbilical cord perivascular cells (HUCPVCs) [10,28]. In addition, iPSCs can be differentiated into MSCs to obtain the quantities required for clinical use [13,14].

It is widely accepted that MSCs exert their capacity to repair and regenerate injured tissues by a paracrine mechanism involving soluble factors and EVs [3,20]. Considering that cell therapy employing MSCs has some limitations, new therapeutic approaches based on the use of MSC-derived EVs are emerging as an alternative for regenerative medicine on liver diseases [29]. These new strategies based on the use of extracellular vesicles rely on the potential of MSCs-EVs to recapitulate most of the therapeutic effect of MSCs [30]. Moreover, an additional attribute is that EVs can be engineered to load specific therapeutics factors and used for their delivery to damaged tissues without the concerns of treating a patient with a genetically modified cell. To assess this goal, the actual strategies include modifying the parental MSCs either by culture condition tuning or gene engineering [23,31,32].

One of the main challenges of MSC-EV therapy is to develop manufacturing strategies that combine an optimal MSC source with a method to isolate EVs at clinically relevant quantities. It should be noted that this strategy must conserve EVs’ properties and biological effect [30,33,34]. In this work, we set up a scalable strategy to produce EVs derived from MSCs for liver fibrosis therapy using ion exchange chromatography. According to the International Society for Extracellular Vesicles (ISEV), differential ultracentrifugation (UC) was the most used technique to separate and concentrate EVs due to the small set of reagents required and good reproducibility [35]. However, the conventional UC protocol for EV isolation involves repeated centrifugation steps, which makes scaling up difficult and results in reduced yield. Moreover, this method impairs EV quality due the co-precipitation of protein aggregates among other contaminants [36,37,38,39]. In recent years, other method such as filtration, size exclusion chromatography, and density gradients were applied to isolate EVs of better quality and obtain different EV populations [40,41,42]. Nevertheless, the small volume that can be processed or the low concentration of the final product are some limitations of these techniques [40,42,43,44]. In this study, we validated that EVs derived from HUCPVCs isolated by the chromatography method keep their properties and in vitro anti-fibrotic effects, in comparison with the gold-standard EV isolated by ultracentrifugation. It should be noted that column chromatographic methods by ion exchange offer a simple, scalable, and fast method for EV isolation [26]. Willms, E. et al. reported that EV compositions and functions could vary depending on the implemented purification and isolation method [45]. In our study, we processed up to 100 mL of CM from HUCPVCs and obtained EVs on three elution fractions of 1 mL with high concentration, quality, and biological activity.

One of the main concerns during the development of MSC-based therapies is the selection of the optimal source of MSCs. Considering that EVs mimic the characteristics of their parental cells, the selection of the optimal source of MSCs remains a point to be considered in the production of EVs for clinical purposes; therefore, comparative studies should be carried out [30]. For instance, Gupta et al. found that adipose-tissue- and Wharton-jelly-derived MSC EVs isolated by the UC method have anti-fibrotic potential, but through different underlying mechanisms [46]. In this work, we performed for the first time a comparison of the therapeutic potential of EVs derived from three clinical relevant sources of MSC, applying a scalable purification method. The three types of EVs have a similar phenotype and morphology. We only found that, while EVs derived from HUCPVCs and iMSCs express high levels of CD9, ASC-EVs express low levels of this marker. Most importantly, the three types of EVs significantly reduce HSC activation, ameliorate liver fibrosis, and induce hepatic regeneration. Considering that the three MSC-EVs showed similar therapeutic potential, additional features are required in order to select the MSC type, such as being robust to expand and the potential to be stocked at a large scale, keeping its biological properties [30,34]. Among the clinically relevant sources of MSCs, the umbilical cord emerged as the preferred source of MSCs for clinical trials In particular, HUCPVCs represent a valuable source of cells for MSC-based therapies due to their fast proliferation rate, high donor homogeneity and accessibility, easy ex vivo manipulation, and increased regenerative, migratory, and immunoregulatory capacity [9,11,47,48].

In recent years, the production of engineered EVs to carry and deliver specific molecular cargos has become an attractive area of research to improve the efficacy and potential of EV-based therapies [3,23,49,50]. In our recent work, we demonstrated that EVs isolated from IGF-I-overexpressing HUCPVCs using the ultracentrifugation method are loaded with IGF-I, which enhances their anti-fibrotic potential [25]. In this study, we tested if the chromatography isolation method is also suitable for obtaining engineered MSC-EVs, keeping the specific cargo and quality. Remarkably, we validated that IGF-I-engineered EVs isolated by IEC strongly reduce liver fibrosis and induce hepatic regeneration. Additionally, we confirmed that the proteome of EVs derived from IGF-I-overexpressing HUCPVCs was not significantly modified and that they contained proteins involved in anti-fibrotic activity.

In summary, in this study, we demonstrated that MSCs-EVs can be efficiently isolated by a scalable method based on ion exchange chromatography, while maintaining their quantity, quality, and biological function. Moreover, we showed that EVs derived from engineered MSCs, particularly HUCPVCs, can carry and deliver specific cargos, such as IGF-I. Importantly, our findings suggest that the chromatographic method preserves the therapeutics potential of engineered EVs, making them a promising alternative for the treatment of liver diseases. All in all, this study provides valuable insights into the development of scalable manufacturing strategies to obtain MSC-derived EVs for their potential use as a therapeutic tool for liver fibrosis.

## 4. Material and Methods

### 4.1. Isolation and Culture of ASCs, HUCPVCs, and iMSCs

Adipose-tissue-derived MSCs (ASCs) were isolated from discarded fat from liposuctions, as we previously described [51]. Briefly, lipoaspirated material was washed extensively with sterile phosphate-buffered saline and then treated with 0.075%-type collagenase (Sigma-Aldrich, St. Louis, MO, USA) in PBS for 30 min at 37 °C with agitation. Cells were centrifuged and pellet was plated in complete DMEM low glucose (Life Technologies, Carlsbad, CA, USA) supplemented with 20% FBS (Internegocios S.A., Mercedes, Argentina) and used for different experiments between passages 4 to 6.

HUCPVCs were isolated from umbilical cord obtained from healthy donors at the Hospital Universitario Austral (Pilar, Buenos Aires, Argentina). as we previously described (Protocol approval #12–038) [51]. In brief, umbilical cords were dissected and vessels with their surrounding Warthon’s jelly were pulled out. The perivascular mesenchymal tissue was removed from the vessels and mechanically disrupted. Minced fragments were plated in complete DMEM low glucose/20% FBS. After 7-day incubation, non-adherent cells and minced fragments were removed and adherent HUCPVCs were cultured and used for different experiments at passages 4 to 6. Induced pluripotent stem cell (iPSC)-derived MSCs (iMSCs) were kindly provided by Ph.D. Carlos Luzzani (LIAN-CONICET, Fleni; Argentina). iMSC differentiation protocol and culture condition were descripted in Luzzani et al. [52].

### 4.2. HUCPVCs Adenoviral Transduction

Recombinant adenoviral vector harboring human IGF-I (AdhIGFI) was previously described [25] and Green Fluorescent Protein gene (AdGFP) was kindly provided by Rodolfo G. Goya (Universidad Nacional de La Plata, La Plata, Buenos Aires, Argentina). HUCPVCs were seeded at 70% of confluence in complete medium. Medium was then removed and cells were infected with AdhIGFI or AdGFP at a multiplicity of infection (MOI) of 1.25, 2.5, 5, 10, 20, or 30 in DMEM low glucose (Life Technologies, USA) and 2% FBS in half of total volume for 2 h. Then, medium was completed with 10% FBS in DMEM low glucose. Non-infected HUCPVCs were also processed as additional control. To set up HUCPVC adenoviral transduction, IGF-I production on CM, cell viability, and anti-inflammatory capacity was evaluated after 3 days of infection (see below).

### 4.3. Conditioned Media (CM) Preparation for EV Isolation

To isolate EVs derived from different MSC sources, ASCs, HUCPVCs, and iMSCs were seeded at 70% of confluence in complete medium. The next day, cells were washed twice with PBS and culture media replaced for DMEM without FBS and phenol red (Life Technologies, USA). Cell supernatants were collected 48 h later and centrifuged at 2500× *g* for 10 min at 4 °C to remove cell debris and to generate conditioned medium (CM).

To isolate EVs derived from AdhIGFI-HUCPVCs, two days after infection, HUCPVCs, AdhIGFI-HUCPVCs, and AdGFP-HUCPVCs were washed and serum-starved. Cell supernatants were collected 48 h later and centrifuged at 2500× *g* for 10 min at 4 °C to remove cell debris and to generate CM.

### 4.4. EV Isolation from Conditioned Media of MSCs

EVs were isolated from CM by anion exchange chromatography using a protocol adapted from Kim et al. [26] Briefly, the chromatography was performed using Q-Sepharose Fast Flow resin (GE Healthcare, Chicago, IL, USA) packed in a column with 4 mL of bed (1 column volume, CV). First, the resin had been equilibrated with 10 CV of 50 mM NaCl in 50 mM phosphate buffer (pH 7.5). Then, the CM was applied directly at room temperature and then washed with 2.5 CV of 100 mM NaCl in 50 mM phosphate buffer (pH 7.5). Finally, elution of retained EVs was performed with sequential application of 2 CV of the 500 mM NaCl in 50 mM phosphate buffer (pH 7.5). The elution fractions obtained denoted as F1, F2, F3, F4, F5, F6, F7, and F8 were stored at −20 °C. EV presence in the fractions was confirmed by protein quantification and flow cytometry. Fractions with EVs were pooled for in vitro and in vivo experiments. Differential ultracentrifugation methods to isolate EVs were described previously [25].

### 4.5. In Vivo Experimental Design: Hepatic Fibrosis Model and Therapeutic Effects of EVs

Six-to-eight-week-old male BALB/c mice were purchased from CNEA (Comisión Nacional de Energía Atómica, Ezeiza, Buenos Aires, Argentina). Animals were maintained at our Animal Resources Facility (Facultad de Ciencias Biomédicas, Universidad Austral) in accordance with the experimental ethical committee and the NIH guidelines on the ethical use of animals. Fibrosis was induced by intraperitoneal (i.p.) administration of 0.2 mg/g bodyweight of thioacetamide (TAA) (Sigma-Aldrich, MO, USA), 3 times per week, for 8 weeks. On week 6, animals were intravenously (i.v.) injected into the tail vein with saline as untreated group or different types of EVs (15 μg/animal/dose), every 5 days for a total of 3 doses. On week 8, animals were euthanized, and liver samples were dissected out, and used for subsequent studies. Independent in vivo experiments were performed to evaluate the therapeutic effects of EVs derived from three different MSC sources (ASC-EVs, HUCPVC-EVs, and iMSC-EVs) or EVs derived from AdhIGFI-HUCPVCs and AdGFP-HUCPVCs as a control. Three independent in vivo experiments were performed (n = 5/6 animals per group).

### 4.6. In Vitro Hepatic Stellate Cell Assay

The CFSC-2G hepatic stellate cell line, originally established from a cirrhotic rat liver, was kindly provided by Dr. Marcos Rojkind (Albert Einstein College of Medicine, New York, NY, USA). Cells were cultured in DMEM (Life Technologies, CA, USA) supplemented with 10% FBS (Life Technologies, Carlsbad, CA, USA) and non-essential amino acids. To evaluate the effect on the activity of HSCs, CFSC-2G cells were incubated with EVs (1 μg/mL) from ASCs, iMSCs, HUCPVCs, AdGFP-HUCPVCs, or AdhIGFI-HUCPVCs for 18 h. Then, cells were washed and collected with Trizol reagent (Sigma-Aldrich, St. Louis, MO, USA) for RNA extraction. Levels of α-SMA and COL1A2 mRNA expression were determined by qPCR.

### 4.7. Ethics Statement

Animals were maintained at our Animal Resource Facilities (School of Biomedical Sciences, Austral University) in accordance with the experimental ethical committee and the NIH guidelines on the ethical use of animals. The “Animal Care Committee” from School of Biomedical Sciences, Austral University, approved the experimental protocol (#2018-08, approval date: 10 August 2018). HUCPVCs and ASCs were obtained from umbilical cord and adipose tissue of healthy donors after informed consent and protocol was approved by the “Institutional Evaluation Committee” (CIE) from School of Biomedical Sciences, Austral University (Protocol No. 16-038, approval date: 18 February 2019).

### 4.8. Statistical Analyses

Data are expressed as mean ± SEM. Statistical analyses were performed using Mann–Whitney test, ANOVA, and Tukey’s post-test, or Kruskal–Wallis and Dunn’s post-test according to data distribution. Data distributions were analyzed by D’Agostino–Pearson omnibus normality test. Differences were significant when *p* < 0.05. All experiments were analyzed in GraphPad Prism Software (version 9.5.0).

Complementary Material and Methods are in the Appendix A.

## Figures and Tables

**Figure 1 ijms-24-09586-f001:**
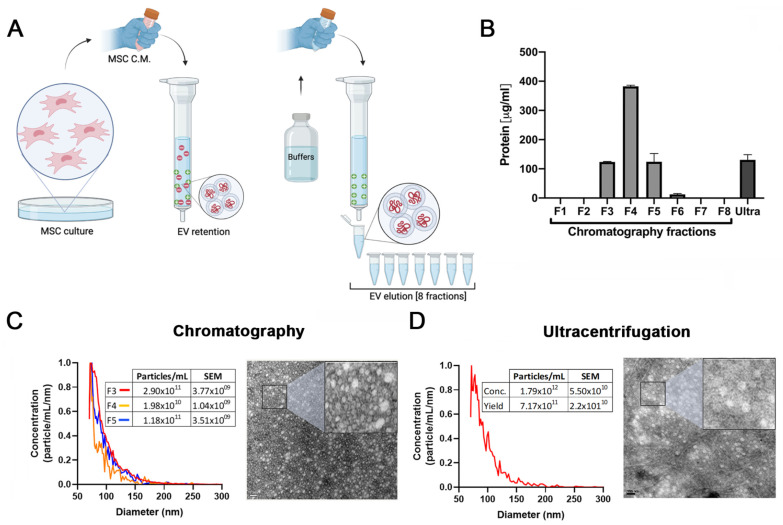
Validation of ion exchange chromatography method to isolate EVs. (**A**) Scheme of EV isolation by ion exchange chromatography. HUCPVCs were incubated for 48 h in Fetal Bovine Serum (FBS)-free medium and the cell culture supernatant collected. Supernatant was centrifuged at 5000 rpm to discard the cellular debris and then applied directly to a column containing the anion exchange resin. After the total volume of HUCPVC supernatant crosses the resin, negatively charged EVs (red “-“ symbols) were retained on the positive charged resin (green “+” symbol). Then, the column was washed and eluted in 8 fractions of 1 mL elution buffer solution. (**B**) Protein quantification by BCA assays of eluted chromatography fractions. (**C**,**D**) EV characterization by MRPS and transmission electron microscopy. Left panel: graph showing quantification and size distribution analysis of fractions #3, #4, and #5 of chromatography and ultracentrifugation pellet assessed by MRPS. Right panel: electron microphotography (scale bar = 100 nm) of HUCPVC-derived EVs isolated by ion exchange chromatography and ultracentrifugation, respectively. (**E**) Confirmation of EV presence on eluted fraction by flow cytometry for CD9 and CD81 EV markers. EVs isolated by ultracentrifugation were used for comparison. Preparation of samples was carried out by trapping EVs with CD63-antibody-coated beads and incubated with specific antibodies conjugated with PE (red line histograms). Beads alone were used as control (black histogram). Graphs show 1 of 3 independent experiments each performed in duplicate. (**F**,**G**) In vitro analysis of EV biological function for liver fibrosis therapy. Hepatic stellate cells (CFSC-2G cell line) were incubated with fractions #3, #4, or #5, a pool of fraction 3–4–5, or EVs isolated by ultracentrifugation (1 µg/mL). DMEM was used as untreated control. After 18 h of incubation, mRNA expression levels of COL1A2 and α-SMA was evaluated by qPCR. Graph shows average of 3 independent experiments performed in triplicate. * *p* < 0.01; ** *p* < 0.001; **** *p* < 0.0001; vs. saline (ANOVA and Tukey’s post-test).

**Figure 2 ijms-24-09586-f002:**
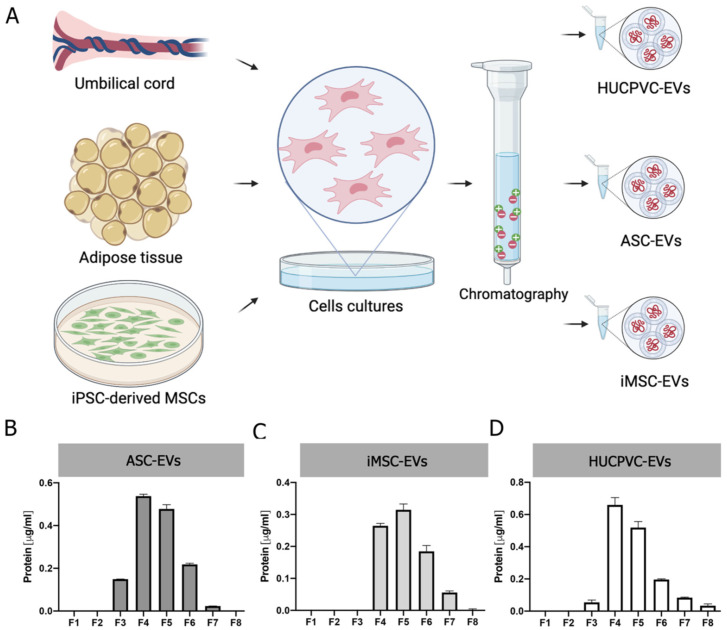
Isolation and characterization of EVs derived from MSCs. (**A**) Scheme of protocol for EV production from human umbilical cord perivascular cells (HUCPVCs-EVs), adipose tissue MSCs (ASC-EVs), and iPSC-derived MSCs (iMSCs-EVs). EVs were isolated by ion exchange chromatography from MSC supernatants after 48 h of culture in fetal-bovine-serum-deprived media. Negatively charged EVs (red “-“symbols), positive charged resin (green “+” symbol). (**B**–**D**) Protein quantification by BCA assays of eluted chromatography fractions on the ASC-EV, iPSC-EV, and HUCPVC-EV isolation. (**E**) Histogram of CD9 and CD81 EV marker analysis by flow cytometry. Preparation of the ASC-EVs, iMSC-EVs, and HUCPVC-EVs was carried out by trapping EVs with CD63-antibody-coated beads and incubated with specific antibodies conjugated with PE (red line histograms). Beads alone were uses as control (black histogram). Graphs show 1 of 3 independent experiments each performed in duplicate. (**F**) EV quantification and size distribution analysis assessed by MRPS.

**Figure 3 ijms-24-09586-f003:**
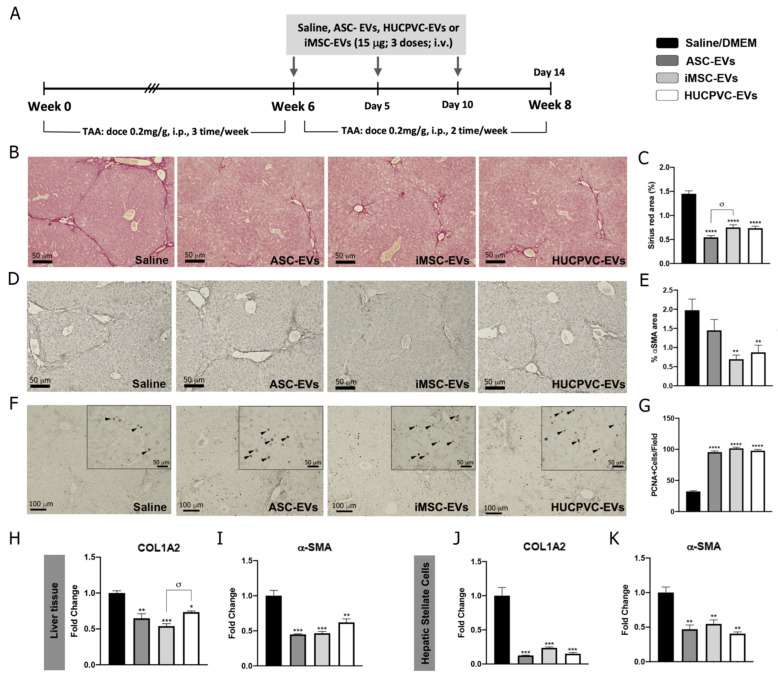
MSC-EV treatment from different sources ameliorates liver fibrosis and promotes hepatic regeneration. (**A**) Experimental design of EV therapy with ASC-EVs, iMSC-EVs, and HUCPVC-EVs. Saline solution was used as vehicle control. Liver fibrosis was induced by TAA administration for 8 weeks (200 mg/Kg/dose, 3 dose/week). At week 6, EVs or vehicle were i.v. administered every 5 days, 15 μg/animal/dose for a total of 3 doses. Animals were euthanized at week 8. Analysis of liver fibrosis by Sirius Red staining: (**B**) representative images of stained liver section (scale bars: 100 mm) and (**C**) morphometric quantification of collagen deposits. Analysis of in vivo HSC activation by α-SMA immunostaining: (**D**) representative images of stained liver sections (scale bars: 100 mm), and (**E**) morphometric quantification of α-SMA-positive area. Analysis of liver regeneration by PCNA immunostaining: (**F**) representative images of stained liver sections (scale bars: 100 mm), squares show 4× amplified images (with PCNA-positive cells indicated by arrowheads), and (**G**) PCNA-positive cell quantification (n = 10 for each group). (**H**,**I**) Analysis of COL1A2 and α-SMA mRNA levels in liver sample of mice 14 days after first dose of EVs. (**J**,**K**) In vitro analysis of COL1A2 and α-SMA mRNA expression on hepatic stellate cells (CFSC-2G cell line) 18 h after EV treatment (1 µg/mL). DMEM was used as untreated control. Graph shows average of 3 independent experiments performed in triplicate. * *p* < 0.05; ** *p* < 0.01; *** *p* < 0.001; **** *p* < 0.0001; * vs. saline; σ vs. iMSC-EVs (ANOVA and Tukey’s post test). Saline/DMEM (black bars), ASC-EVs (dark gray bars), iPSC-EVs (light gray bars), and HUCPVC-EVs (white bars).

**Figure 4 ijms-24-09586-f004:**
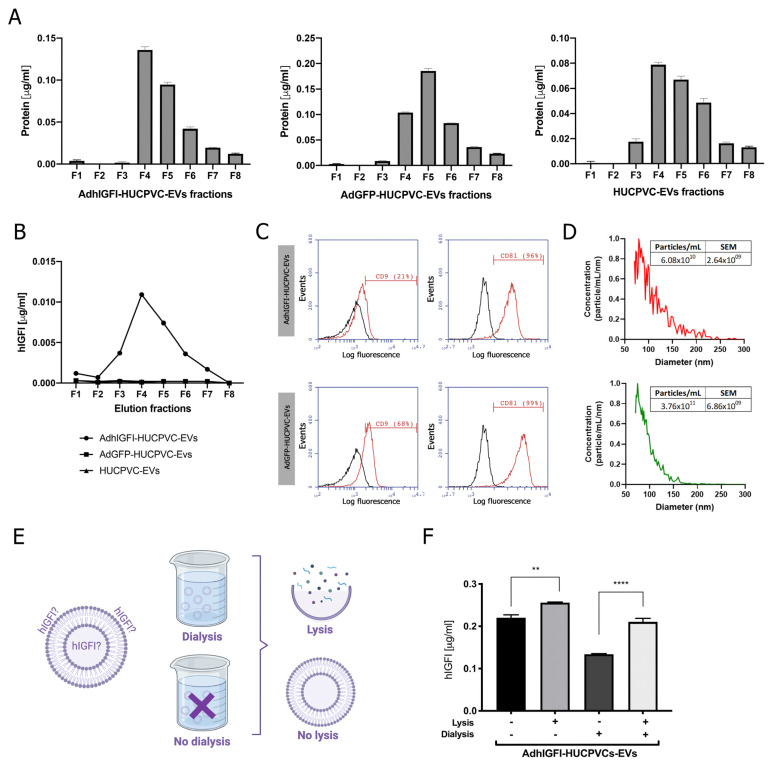
EVs derived from AdhIGFI-HUCPVCs retain the specific cargo of IGF-I after being isolated by chromatography. (**A**) Protein quantification by BCA assays of eluted chromatography fractions on the AdhIGFI-HUCPVC-EV, AdGFP-HUCPVC-EV, and HUCPVC-EV isolation. (**B**) IGF-I quantification by ELISA on eluted chromatography fractions. (**C**) Histogram of CD9 and CD81 EV marker analysis by flow cytometry. Preparation was carried out by trapping EVs with CD63-antibody-coated beads and incubated with specific antibodies conjugated with PE (red line histograms) and beads alone as control (black lines histogram). (**D**) EV quantification and size distribution analysis assessed by MRPS. (**E**) Experimental design of EV processing to determine IGF-I localization. EVs isolated by chomatography were dialyzed on 300 kDa membrane against PBS and lysated on lysis buffer. (**F**) Dosage of IGF-I in EVs derived from AdhIGFI-HUCPVCs, lysated (gray bars) or non-lysated (black bars), with or without dialysis, determined by ELISA. Increased IGF-I levels were observed on lysed–dialyzed AdhIGFI-HUCPVC-EVs; **** *p* < 0.0001; ** *p* < 0.01 vs. lysed condition (ANOVA and Tukey’s post-test). Graph shows average of 3 independent experiments performed in triplicate each.

**Figure 5 ijms-24-09586-f005:**
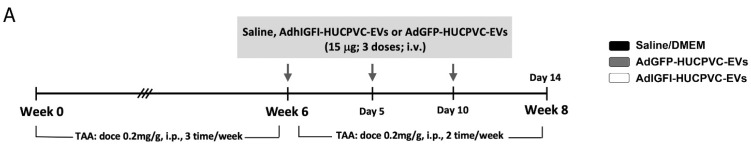
AdhIGFI-HUCPVCs-derived EV treatment isolated by chromatography ameliorates liver fibrosis. (**A**) Experimental design. AdhIGFI-HUCPVC-EVs, AdGFP-HUCPVC-EVs (15 μg/animal/dose, every 5 days for a total of 3 doses), or vehicle were administrated after 6 weeks of liver fibrosis induction by TAA administration. Animals were euthanized at week 8. Analysis of liver fibrosis by Sirius Red staining: (**B**) representative photomicrographs of stained liver section (scale bars: 100 mm) and (**C**) morphometric quantification of collagen deposits. Analysis of in vivo HSC activation by α-SMA immunostaining: (**D**) representative photomicrographs of stained liver sections (scale bars: 100 mm), and (**E**) morphometric quantification of α-SMA positive area. Analysis of liver regeneration by PCNA immunostaining: (**F**) representative photomicrographs of stained liver sections (scale bars: 100 mm), squares show 4× amplified images (with PCNA-positive cells indicated by arrowheads), and (**G**) PCNA-positive cell quantification (n = 10 for each group). (**H**,**I**) Analysis of COL1A2 and α-SMA mRNA levels in liver sample of mice 14 days after first dose of EVs. (**J**,**K**) In vitro analysis of COL1A2 and α-SMA mRNA expression on hepatic stellate cells (CFSC-2G cell line) 18 h after incubation with EVs (1 µg/mL). DMEM was used as untreated control. Graph shows average of 3 independent experiments performed in triplicate. Saline/DMEM (white bars), AdGFP-HUCPVC-EVs (gray bars), or AdhIGFI-HUCPVC-EVs (black bars) administration. * *p* < 0.05; ** ^λλ^
*p* < 0.001; *** *p* < 0.0005, **** ^λλλλ^
*p* < 0.0001; * vs. saline; ^λ^ vs. AdGFP-HUCPVC-EVs; (ANOVA and Tukey’s post-test).

**Figure 6 ijms-24-09586-f006:**
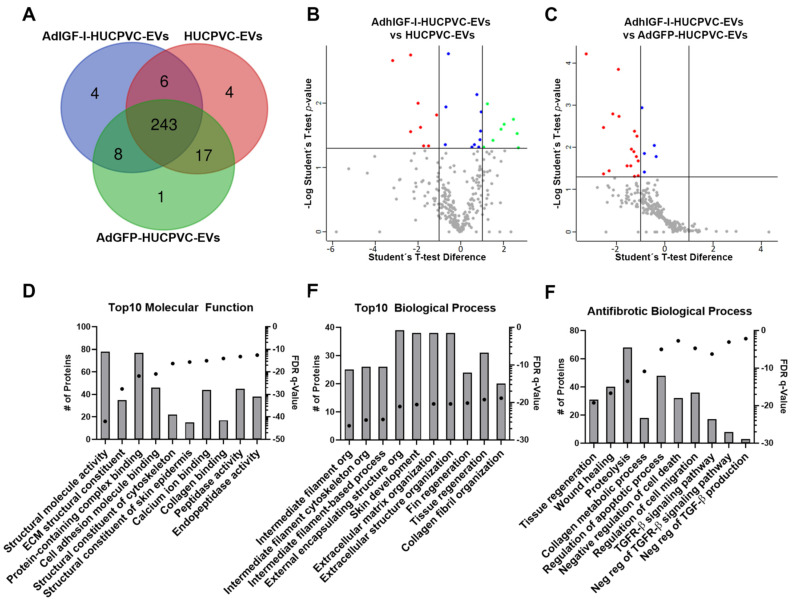
The proteome of EVs derived from HUCPVCs is related with anti-fibrosis potential. (**A**) Venn diagram of proteins identified by LC-MS/MS on AdhIGF-I-HUCPVC-EVs, AdGFP-HUCPVC-EVs, and HUCPVC-EVs. Volcano plot showing the differential expression analysis between AdhIGF-I-HUCPVC-EVs and HUCPVC-EVs (**B**), or AdGFP-HUCPVC-EVs (**C**). Red dots and green dots represent downregulated (*p* < 0.05, fold change < 0.05) and upregulated (*p* < 0.05, fold change > 2) proteins respectively. Blue dots represent significative non-regulated proteins (*p* < 0.05, 0.05 < fold change < 1). Gray dots represents non-significative proteins (*p* > 0.05). (**D**–**F**) Gene ontology analysis of co-expressed proteins on AdhIGF-I-HUCPVC-EVs, AdGFP-HUCPVC-EVs, and HUCPVC-EVs. (**D**,**E**) show the top 10 of “Molecular Function” and “Biological Function”, respectively. (**F**) show the “Biological Process” involved in anti-fibrotic pathways. Graph shows number of proteins (right axis, column bars) and Q-value B&H (left axis, dots). FDR, false discovery rate.

## Data Availability

Appendix A is available.

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
