# Peer review of "Chromatographic Scalable Method to Isolate Engineered Extracellular Vesicles Derived from Mesenchymal Stem Cells for the Treatment of Liver Fibrosis in Mice"

_ijms, 2023, doi:10.3390/ijms24119586_

Round 1

Reviewer 1 Report

The manuscript Chromatographic Scalable Method to Isolate Engineered Extracellular Vesicles Derived from Mesenchymal Stem Cells for the Treatment of Liver Fibrosis in Mice describes new method of purification of Extracellular Vesicles and tests the potential therapeutic interest of these EV.

The paper is well written and is of potential very high interest. I recommand its publication.

Bellow are indicated some minor points:

In abstract, remove "urgently". The urgence notion is relative, not absolute.

Page 2, introduction:

The references 2 and 3 should be separated by a comma.

"lower donor variability". Please explain which variability? How is it measured? Not clear.

Figure 1, label c: Chromatography, with h

Section 2.6. Why a cap H in Hepatic?

Section 2.7: the use of p<0.05 is clearly not enough stringent. For example take a look at :

Colquhoun, D., 2014. An investigation of the false discovery rate and the misinterpretation of p-values. Royal Society Open Science 1, 140216-140216.

"If you use p = 0.05 to suggest that you have made a discovery, you will be wrong at least 30% of the time."

Johnson, V.E., 2013. Revised standards for statistical evidence. Proceedings of the National Academy of Sciences of the United States of America 110, 19313–19317.

"An examination of these connections suggest that recent concerns over the lack of reproducibility of scientific studies can be attributed largely to the conduct of significance tests at unjustifiably high levels of significance. To correct this problem, evidence thresholds required for the declaration of a significant finding should be increased to 25–50:1, and to 100–200:1 for the declaration of a highly significant finding. In terms of classical hypothesis tests, these evidence standards mandate the conduct of tests at the 0.005 or 0.001 level of significance."

Section 3.1: "present similar phenotype". Rather you should say that you do not detect difference.

Legend of figure 4: "done in triplicate" change to "done in triplicate each".

Reviewer 2 Report

Submitted manuscript is very interesting and it might be a good addition to the SI Mesenchymal Stem Cells: Immunobiology and Role in Immunomodulation and Tissue Regeneration 2.0. However, the writing style is a bit to specific for interdisciplinary journal and wide audience. It would be nice to add short description of multipotent mesenchymal stromal cells and their ability to differentiate in vitro into theosteogenic, chondrogenic, and adipogenic lineages etc. (Cell StemCell16 (2015) 239; Sensors 2017, 17(11), 2605, etc. ).

Manuscript has clear technical deficiency and in the present state it is not useful for the other researches. Namely – the majority of figures are represented at such a scale that they do not work even as simple indicators – there is no chance to understand the meaning as al symbols are too small. For example, Figure 5 F has insets, what was the scale for it? Figure 6 has writings (D,E,F) which are not readable. One could increase the font size or write 1,2,3, … and make notes in the figure caption.

The rule is that all important details must be given using the size comparable with the main font.

Reviewer 3 Report

The manuscript “Chromatographic Scalable Method to Isolate Engineered Extracellular Vesicles Derived from Mesenchymal Stem Cells for the Treatment of Liver Fibrosis in Mice” fits the SI scope. The authors present the Mesenchymal stem cells (MSC)-derived extracellular vesicles (EVs) obtaining and separation, followed by evaluation using in vivo and in vitro biological assays. The methodology and the design of the experiments is well explained, and presented graphically. The quality of presentation is good, and the conclusions are based on the results. Before publication, some minor issues should be corrected/justified:

In future works, please insert the line numbering

Please correct minor English errors throughout the manuscript (section 2.4, 4)

Section 2.7 please add the software's version

Section 2.5

Please explain the use of the same dose to all mice, independent of their weight

Please provide the information regarding the bioethical approve letter

Please add a list with the independent in vivo experiments (at least the information presented in figure 5A)

Also, please add in this section the number of animals in each group
